# Epidemiology, Risk Factors, and Outcomes of Opportunistic Infections after Kidney Allograft Transplantation in the Era of Modern Immunosuppression: A Monocentric Cohort Study

**DOI:** 10.3390/jcm8050594

**Published:** 2019-04-30

**Authors:** Philippe Attias, Giovanna Melica, David Boutboul, Nathalie De Castro, Vincent Audard, Thomas Stehlé, Géraldine Gaube, Slim Fourati, Françoise Botterel, Vincent Fihman, Etienne Audureau, Philippe Grimbert, Marie Matignon

**Affiliations:** 1AP-HP (Assistance Publique-Hôpitaux de Paris), Nephrology and Renal Transplantation Department, Groupe Hospitalier Henri-Mondor/Albert-Chenevier, 94010 Créteil, France; philippe.attias@aphp.fr (P.A.); vincent.audard@aphp.fr (V.A.); thomas.stehle@aphp.fr (T.S.); philippe.grimbert@aphp.fr (P.G.); 2AP-HP (Assistance Publique-Hôpitaux de Paris), Infectious Disease Department, Groupe Hospitalier Henri-Mondor/Albert Chenevier, 94010 Créteil, France; giovanna.melica@aphp.fr (G.M.); geraldine.gaube@aphp.fr (G.G.); 3AP-HP (Assistance Publique-Hôpitaux de Paris), Clinical Immunology Department, Hôpital Saint Louis, 75010 Paris, France; david.boutboul@aphp.fr; 4INSERM U967 HIPI, Université Paris Diderot, 75012 Paris, France; 5AP-HP (Assistance Publique-Hôpitaux de Paris), Infectious Disease Department, Hôpital Saint Louis, 75010 Paris, France; nathalie.de-castro@aphp.fr; 6Université Paris-Est-Créteil, (UPEC), DHU (Département Hospitalo-Universitaire) VIC (Virus-Immunité-Cancer), IMRB (Institut Mondor de Recherche Biomédicale), Equipe 21, INSERM U 955, 94010 Créteil, France; 7AP-HP (Assistance Publique-Hôpitaux de Paris, Clinical Microbiology Department, Virology, Bacteriology and Infection Control Units, 94010 Créteil, France; slim.fourati@aphp.fr (S.F.); francoise.botterel@aphp.fr (F.B.); vincent.fihman@aphp.fr (V.F.); 8Université Paris-Est-Créteil (UPEC), DHU (département hospitalo-universitaire) VIC (virologie immunité cancer), IMRB (Institut Mondor de Recherche Biomédicale), INSERM U955, équipe 18, 94010 Créteil, France; 9EA Dynamyc, Université Paris Est Créteil– Ecole vétérinaire de Maison Alfort, F-94000 Créteil, France; 10Université Paris-Est-Créteil, UPEC, DHU (Département Hospitalo-Universitaire) A-TVB, IMRB (Institut Mondor de Recherche Biomédicale)- EA 7376 CEpiA (Clinical Epidemiology And Ageing Unit), 94010 Créteil, France; etienne.audureau@aphp.fr; 11AP-HP, CIC-BT 504, 94010 Créteil, France

**Keywords:** kidney transplantation, opportunistic infection, allograft survival, BK virus nephropathy

## Abstract

Epidemiology of opportunistic infections (OI) after kidney allograft transplantation in the modern era of immunosuppression and the use of OI prevention strategies are poorly described. We retrospectively analyzed a single-center cohort on kidney allograft adult recipients transplanted between January 2008 and December 2013. The control group included all kidney recipients transplanted in the same period, but with no OI. We analyzed 538 kidney transplantations (538 patients). The proportion of OI was 15% (80 and 72 patients). OI occurred 12.8 (6.0–31.2) months after transplantation. Viruses were the leading cause (*n* = 54, (10%)), followed by fungal (*n* = 15 (3%)), parasitic (*n* = 6 (1%)), and bacterial (*n* = 5 (0.9%)) infections. Independent risk factors for OI were extended criteria donor (2.53 (1.48–4.31), *p* = 0.0007) and BK viremia (6.38 (3.62–11.23), *p* < 0.0001). High blood lymphocyte count at the time of transplantation was an independent protective factor (0.60 (0.38–0.94), *p* = 0.026). OI was an independent risk factor for allograft loss (2.53 (1.29–4.95), *p* = 0.007) but not for patient survival. Post-kidney transplantation OIs were mostly viral and occurred beyond one year after transplantation. Pre-transplantation lymphopenia and extended criteria donor are independent risk factors for OI, unlike induction therapy, hence the need to adjust immunosuppressive regimens to such transplant candidates.

## 1. Introduction

Kidney allograft recipients are exposed to a broad range of infectious pathogens that give rise to infections with unusual and more severe presentations [1]. Opportunistic infections (OIs) include infections caused by uncommon pathogens and those caused by common pathogens but with unusual and more severe forms [2]. The reported incidence of OIs is variable, from 10% to 25% [3,4]. Currently, prevention strategies against cytomegalovirus (CMV), herpes simplex viruses (HSV), and *Pneumocystis* spp. are recommended and result in a significant reduction of post-transplantation OIs [5] and 50% decrease in the risk of death due to infectious causes. However, infections remain the most common cause of non-cardiovascular deaths (15–20%) [5,6]. 

After solid-organ transplantation (SOT), OIs flourish in the first 12 months boosted by the immunosuppressive status [2] since less than 20% of SOT recipients receive no induction therapy and up to 60% of kidney transplant recipients receive a T-cell depleting agent [7,8]. Anti-thymocyte globulin primarily induces rapid, profound, and long-lasting depletion of T-lymphocytes in peripheral blood and lymphoid organs, and apparently it does not spare B-cell and NK cell populations [9,10]. Thanks to such therapies, patient and kidney allograft survival after kidney transplantation have markedly improved and acute allograft rejection has decreased [11,12,13]. On the other hand, one could argue that the long duration of immunosuppression might be the culprit for the increased incidence of OIs. 

The epidemiology of OIs after SOT was previously described in two large cohorts on transplant recipients. The first one was conducted 10 years ago and included SOT recipients treated with alemtuzumab [4]. They showed that receiving lung or intestinal transplants was independent risk factors for OIs [4]. Published in the era of modern immunosuppression and after the wide use of prevention strategies, the second study included abdominal SOT recipients (kidney, pancreas, and liver), hence the heterogeneous patient profiles and immunosuppressive regimens [3]. The authors highlighted the delayed onset of OIs where most infections occurred after six months without any impact on recipient’s survival and graft function [3]. A recent pediatric cohort on kidney allograft recipients has confirmed the absence of impact of viral OIs (CMV, Epstein Barr virus (EBV), and BK virus (BKV)) on kidney allograft survival [14]. In other studies on kidney allograft recipients, only selected OIs, secondary to specific pathogens (*Nocardia, Aspergillus, Cryptococcus neoformans*), have been reported [15,16,17].

Given the lack of clinical and epidemiological data on OIs after kidney allograft transplantation, we conducted a large monocentric cohort study on all kidney allograft recipients in our center to analyze the epidemiology of OIs and their impact on kidney recipient survival and allograft function. 

## 2. Materials and Methods

### 2.1. Study Design and Patients

We conducted a single center retrospective cohort enrolling all adult kidney allograft recipients registered between January 2008 and December 2013. We excluded cases with primary allograft non-function happening within seven days after transplantation. Expanded criteria donor (ECD) was defined as donors older than 60 years or between 50 and 60 years, with two of the three following criteria: (i) hypertension; (ii) pre-retrieval serum creatinine > 1.50 mg/dL; and (iii) cerebrovascular cause of brain death [18]. Glomerular filtration rate was estimated (eGFR) using MDRD formula [19]. Acute rejection episodes were classified according to updated Banff classification [20]. Allograft loss was considered if eGFR was below 15 mL/min/1.73 m^2^. All recipients were followed at least one year after transplantation unless death or graft loss occurred earlier.

### 2.2. Infectious Prophylaxis

The management for CMV prophylaxis followed international recommendations [21]. Prophylaxis involved the administration of oral valganciclovir to high (D+/R-) and intermediate (R+ treated with thymoglobulin) risk patients. Duration of prophylaxis was 6 months in high risk patients and 3 months in intermediate ones.

Participants with past history of tuberculosis were treated with isoniazid for three months after transplantation. *Pneumocystis jirovecii* prophylaxis included trimethoprim-sulfamethoxazole (400 mg) or pentacarinat aerosol for 12 months after transplantation and till CD4 count dropped to <200/µL. 

### 2.3. Opportunistic Infections

OIs were defined according to current literature [1] and international guidelines [22,23]. All episodes were retrospectively and blindly validated (review of all medical reports without the patient name and the final conclusion (clinical and biological data) of infections that happened in kidney-transplant recipients included in the study) by an infectious disease specialist part of the study group. The following OIs were considered:

-Bacteria: *Mycobacterium* sp., *Listeria monocytogenes* and *Nocardia* sp. 

-Virus: CMV, active replication of HSV, Varicella-Zoster virus (VZV), Human Herpes Virus-8 (HHV8), BKV, Norovirus, and JC virus.

We included BKV infection, as BK virus, highly seroprevalent in humans, appears to cause clinical disease only in immunocompromised patients and almost all after kidney transplantation (tubulointerstitial nephritis called BKV-induced nephropathy directly related to plasma viral load) [24]. In our center, during the first year after kidney transplantation, BK viruria tests were performed at 1, 2, 3, 6, 9, and 12 months. BK viremia was checked once BK viruria was positive. If BK viruria (associated with BK viremia or not) was positive, a blood test was performed every two weeks.

We also considered Kaposi sarcoma, as one of the four types was organ transplant-associated and usually regresses with reduction in immunosuppression [25]. 

-Fungi: Candida spp, Cryptococcus spp., invasive molds, and *Pneumocystis jirovecii.*


-Parasites: Toxoplasma gondii, Microsporidium sp, Cryptosporidium sp, Leishmania sp.

### 2.4. Endpoints

Clinical endpoints were an OI episode, death, and allograft loss. Recipients with at least one episode of OI were compared with the control group which included all other kidney allograft recipients engrafted at the same time period.

### 2.5. Statistical Analysis

Continuous variables are presented as mean (± Standard Deviation (SD)) or median (Interquartile Range (IQR)). Categorical variables are presented as counts (%). Baseline donor, recipient, and kidney transplant characteristics were compared between OI and control groups using Student *t*-test or Wilcoxon test for continuous variables, and Chi-2 or Fisher’s exact tests for categorical variables, as appropriate. Time-to-event survival analyses were conducted to determine predictors of OI occurrence, patient overall survival, and allograft survival. Survival curves were plotted using Kaplan–Meier method and logrank tests to assess significance upon group comparison. Time varying Cox proportional hazard models were built for each endpoint, and hazard ratios (HR) along with their 95% confidence intervals (95% CI) were calculated. Factors yielding *p* < 0.2 in the univariate analyses were then considered in the multivariate analyses’ models, using a stepwise backward approach by sequentially removing variables not significant at *p* < 0.1 until the final model was reached. Variables with available repeated data over time were entered both as time-fixed (value at the time of transplantation) and as time-varying (all available time points) variables into the Cox model. No imputation of missing data was done. Competing risk survival analysis (e.g., Fine–Gray methodology) cannot be directly applied on time-varying variables, therefore only results from Cox models are reported for allograft survival. All tests were two-tailed, and the significance level was reached with *p* value < 0.05. The analysis was performed using Stata SE v15.1 (College Station, TX, USA). 

## 3. Results

### 3.1. Whole Cohort

A flow-chart of the study population is presented in Figure 1. Between January 2008 and December 2013, 557 kidney transplantations were performed in 557 patients (*n*), of whom 19 showed early primary allograft non-function. Overall, only 538 transplantations in 538 patients were included. Mean age was 52 ± 14 years. Mean follow-up was 55 ± 24 months. At the end of follow-up period, patient survival was 88% with 65 deaths, allograft survival was 87% with 72 allograft losses, and mean eGFR was 48 ± 20 mL/min/1.73 m^2^. Table 1 and Table 2 described the whole cohort.

### 3.2. Opportunistic Infections 

Eighty OI episodes were reported in 15% of patients (*n* = 72). The median time to post-transplantation OI was 12.8 (6.0–31.2) months, and in 39 patients (48.8%), OI occurred over the first post-transplantation year.

Viruses were the leading cause of OI episodes, *n* = 54 (68%), representing 10% of the whole cohort. Median time to viral OI onset was 14 (7–31) months after transplantation. Of those viral OIs, we recorded 21 (39%) shingles (4%-whole cohort), 18 (33%) BKV nephropathy (BKVN) (3%-whole cohort), 6 (11%) Kaposi sarcoma (1%-whole cohort), 3 (6%) CMV disease (0.5%-whole cohort), 3 (6%) norovirus gastroenteritis (0.5%-whole cohort), and 1 (2%) of each of the following: JC virus causing progressive multifocal leukoencephalopathy (PML) (0.2%-whole cohort), VZV retinitis (0.2%-whole cohort), and HSV-1 esophagitis (0.2%-whole cohort). 

Fungal infections were the second most common OIs, registered in 15 patients (19%) (3%-whole cohort), in the first 6 (2–25) months after transplantation, which is significantly earlier than viral infections (*p* = 0.04). We counted five (33%) invasive candidiasis (0.9%-whole cohort), four (27%) invasive aspergillosis (IA) (0.7%-whole cohort), three (20%) cryptococcosis (0.5%-whole cohort), two (13%) *Pneumocystosis* pneumonia (PCP) (0.3%-whole cohort), and one (7%) disseminated *Trichophyton Rubrum* infection (0.2%-whole cohort).

Among the six (7%) parasitic infections (1%-whole cohort) occurring 16 (5–23) months after transplantation, four were cryptosporidiosis (0.7%-whole cohort) and two microsporidiosis with gastrointestinal involvement (0.3%-whole cohort). Finally, five (6%) bacterial infections (0.9%-whole cohort) were described, of which two (40%) were tuberculosis (0.3%-whole cohort), two (40%) were nocardiosis (0.3%-whole cohort), and one (20%) was disseminated atypical mycobacteria infection (0.15%-whole cohort). Time to post-transplantation infection was 11 (9–34) months. Seven (10%) recipients had more than one post-transplantation OI episode. 

The comparison between OI and control groups is shown in Table 1 and Table 2. Donors were significantly older in OI group than in control group (*p* = 0.02), with a similar statistical trend in recipients (*p* = 0.056). At the time of transplantation, blood lymphocytes count was significantly lower in OI group (*p* = 0.04). Numbers and percentages of CD4 and CD8 T-cells were similar in both groups; the same was found for the immunosuppressive treatments after transplantation (induction and maintenance). 

The estimated GFR in OIs group was significantly lower than in control group at any given time (i.e., at 12-months or last available follow-up data). Acute rejection incidence and CMV viremia were similar in both groups. At the end of follow-up, event rates, allograft loss, and time to death after transplantation were similar in both groups. 

In time-to-event analysis, the univariate risk factors for OIs after kidney transplantation (Table 3) were older recipient age (HR 1.02 (1–1.04), *p* = 0.03), older donor age (1.02 (1.01–1.04), *p* = 0.02), and ECD (2.76 (1.68–4.54), *p* < 0.0001). Higher CD4+ T-cells during follow-up and higher blood lymphocyte count at the time of transplantation were protective factors against OI (0.31 (0.11–0.83) and 0.61 (0.40–0.95), respectively). At the time of transplant, blood lymphocytes count was significantly lower in patients with OI (Table 1 (OI) Median 1.2 (IQR 0.9–1.6) vs. (Controls) 1.3 (1.0–1.8); *p* = 0.04) while CD4/CD8 numbers (%) were similar in both groups (Table 1) or using time-to-event analysis (Table 3). Induction and maintenance immunosuppressive regimens, acute rejection episode, and CMV viremia were not OI risk factors.

Independent risk factors for OI according to multivariate analysis were ECD (2.53 (1.48–4.31), *p* = 0.0007), and BK viremia (6.38 (3.62–11.23), *p* < 0.0001). High blood lymphocyte count at the time of transplantation was an independent protective risk factor (0.60 (0.38–0.94), *p* = 0.026). The multivariable analysis conducted only on patients with available pre-transplantation CD4 T-cell counts (*n* = 456) showed that ECD (2.92 (1.62–5.27), *p* = 0.0004) and BK viremia (5.11 (2.72–9.57), *p* < 0.0001) were independent risk factors for OI. In contrast, a higher CD4 T-cell percentage during follow-up (time-varying variable) (0.98 (0.96–0.99), *p* = 0.015) and, to a lesser extent, a higher lymphocyte count at the time of transplantation (0.68 (0.44–1.07), *p* = 0.09) were independent protective factors.

### 3.3. Patients and Allograft Survival

In OI group, patient survival was significantly lower than in control group (Figure 2a, *p* = 0.009). After OI episode, 10 patients (14%) died, of whom three (30%) deaths were related to an OI episode (one PML, one PCP, and one IA). Other causes of death included cardio-vascular disease (*n* = 3), hemorrhagic shock (*n* = 1), traumatism (*n* = 1), bacterial infections (*n* = 1), and neoplasia (*n* = 1). OI was not an independent risk factor for death as shown by the multivariable analysis (Table 4). OI lost its statistical significance after multivariable adjustment for recipient age at transplantation, TCD8 cells (/1000) during follow-up, neutrophils (/1000) during follow-up, HCV+ status, former kidney transplantation and diabetes. Consequently, and in accordance with our statistical analysis strategy (section #2), OI was left out of Table 4 showing only results from the final multivariable model after a stepwise backward approach was applied.

Allograft survival was significantly lower in OI group (Figure 2b, *p* = 0.0002). After OI episode, allograft loss occurred in 13 (18%) patients, around 31 (5–63) months after transplantation. Causes of allograft loss were five (38%) BKVN, five (38%) chronic allograft dysfunction, two (16%) refractory acute rejection, and one (8%) unknown cause. OI episode was an independent risk factor for allograft loss with HR = 2.53 (1.29–4.95) (*p* = 0.007) (Table 4). 

### 3.4. Analysis Excluding BKVN

As BKVN is well-known to cause a chronic destructive infection [24], we performed another analysis excluding BKVN events (Table 3). ECD and low blood lymphocytes count at the time of transplantation remained the two independent risk factors for OI episode (4.09 (2.06–8.09), *p* < 0.0001 and 0.64 (0.38–1.06), *p* = 0.08, respectively). OI was not found to be a risk factor for allograft loss (*p* = 0.87; Figure 2c and Appendix A).

## 4. Discussion

We present here the results of a monocentric cohort analysis conducted on more than 500 kidney allograft recipients. We showed that, in the era of modern immunosuppression and the wide use of infectious disease prophylactic strategies, OIs occurred more than one year after transplantation and that pre-transplantation lymphopenia was an independent risk factor for OI episode, which was not the case for induction therapy. Moreover, OIs were an independent risk factor for allograft loss but had no effect on patient survival. 

Although OIs are well defined in the setting of HIV [23], no classification of post-SOT OIs is currently available [2]. However, we tried in our work to carefully apply the current OI definitions on post-SOT settings taking into account the standardized immunosuppressive regimen and the type of SOT. On this point, former studies on allograft recipients were quite heterogenous concerning the infections considered and the type of SOT [3,4,14]. To our knowledge, no study evaluating the risk factors for OIs versus more severe common infections in engrafted patients has been published. Therefore, no conclusion regarding physiopathology and risk factors is available. In our cohort, we used HIV classification to define OI updated with BKVN, an immunosuppression-induced infection after kidney transplantation [23,24]. This selection process allowed us to provide reliable data on incidence and spectrum of OI after kidney transplantation and could be routinely used by clinicians to customize the prevention strategies to the patient condition. 

OI proportion in our cohort was significantly lower than the most recently published incidence rate of around 25% [3]. Several explanations may account for this low incidence. First, the post-transplantation CMV, PCP, and bacterial prophylaxis strategies we use in our center are in fulfilment with the international recommendations (e.g., trimethoprim-sulfamethoxazole for Nocardia) [5,21]. Secondly, the immunocompromised recipients were exposed to a lower level of CNI, a strategy previously described to significantly decrease OI incidence [26]. At last, solid-organs failure before transplantation induced variable degrees of immune suppression. For instance, liver cirrhosis is associated with dysfunction of the defensive mechanisms against infections and higher incidence of sepsis [27] unlike end-stage renal failure [28]. Therefore, fungal infections risk is lower after kidney transplantation compared with other SOT populations [29]. 

Thus, we updated the description of post- kidney transplantation OIs to align it with the new strategies of immunosuppressive therapy. In our cohort, the incidence of CMV disease was significantly lower than previously described, probably because of the application of the regularly-updated prevention recommendations [2]. However, viral infections remained the first cause of OIs, mainly cutaneous shingles and BKVN. No prevention strategy is currently recommended for shingles. BKVN is clearly problematic after kidney transplantation since it thrives in immune suppression status, has a great impact on kidney allograft survival, and there is no curative treatment for it [24]. IA incidence is also lower in our cohort [29], whereas other OIs incidence was in the previously described range [16] after kidney transplantation.

Interestingly, time to OI onset was long, more than one year after transplantation. The latest review has reported a peak of OI at 6–12 months after transplantation [2]. Again, prevention strategies could probably postpone post-transplantation infections onset. However, post-transplantation fungal infection developed significantly earlier as in former studies, which confirmed that those infections flourish by the peak of immunosuppression [29]. No prevention strategy is currently recommended for those infections as well as PCP.

Thereafter, we aimed to identify independent risk factors for post-kidney transplantation OI. We found that ECD and low pre-transplantation lymphocyte count were independent risk factors; the type of induction immunosuppressive treatments and the recipient age were not. In kidney allograft recipients, older donor age, irrespective of recipient age, increases the rate of acute allograft rejection and infections [30,31]. The underlying immune system seems to be more important than immunosuppressive therapy. Aged transplanted mice could have an impaired anti-infectious response with accumulation of memory CD4+ T-cells and reduced Th1 anti-donor immune response [32,33]. These immunological effects could significantly decrease anti-infectious response in recipients transplanted from ECD. High CD4+ T-cells count was significantly a protective factor, but there was no effect of CD8+ T-cells count while CD4/ CD8 numbers (%) at the time of transplant were similar in both groups. The total count in lymphocytes had a superior predictive value for OI than the separate levels of CD4/CD8. However, the study population for analyses on CD4/CD8 was slightly decreased due to missing information on these variables, thus possibly resulting in a moderate loss of statistical power. High late stage differentiated CD28+CD57+CD4+ T-cells rates at the time of transplantation is independently associated with a decreased risk of OI [28]. Analysis of naive CD4+ T-cells remains to be determined since such phenotype has been associated with a high risk of infection in patients with common variable immunodeficiency [34]. Surprisingly, immunosuppressive induction using depletive monoclonal agents was not associated with OI incidence. Comparing the risk of infection with depletive and non-depletive therapies yielded controversial data although the most recent work shows that thymoglobulin was not associated with higher infection risk [35,36,37]. Almost all of our patients were treated with induction therapy. No induction therapy in immunocompromised kidney allograft recipients could be an option [38]. Whether the absence of induction could be associated with a significantly lower incidence of OI need to be elucidated. 

How lymphopenia before transplant could influence OI occurring more than one year after transplantation remains unknown. Again, the wide use of prophylaxis (trimethoprim-sulfamethoxazole and valganciclovir) prevents early infection (mostly PCP, Nocardia, and CMV disease). Considering late infection, we believed that lymphopenia before transplantation could be a cumulative effect of immunosuppressive therapies in older patients.

Our data confirm that OI is not an independent risk factor for death [3,4]. In a recent large Finnish cohort, OI rarely caused deaths after kidney transplantation, but the most common cause of infection-related mortality was common bacterial infections, e.g. septicemia and pneumonia [6]. The lack of OI-related effect on mortality compared with the role of common bacterial infections needs deeper analyses of causes and risk factors for common infections; this should enable us to adjust prevention strategies to different contexts. Additionally, recent data suggest that infections could be the first cause of death after transplantation [39].

Finally, in our cohort, OI was an independent risk factor for allograft loss only if BKVN episodes were considered. The negative impact of BKVN on kidney allograft survival is well-documented [24]. Thus, in one of the analyses, we excluded BKVN from OI episodes and found no impact on kidney allograft survival on the long-term [3]. To decrease BKVN, only m-TOR inhibitors based immunosuppressive combination showed a significant effect, thus should be considered in all patient with standard immunologic risk [40].

Our study presents limits. The first one is being a single center study and retrospective. These results must be confirmed in a prospective multicentric cohort. However, the single center study implies only one way to manage immunosuppression after transplantation. The second one is that we performed an overview of OI without considering specific prognosis of each infection.

In conclusion, our study showed that, in the era of modern immunosuppression and the wide use of infection prophylactic regimens, OIs occurred later, more than one year after kidney transplantation and were mainly viral. Pre-transplantation lymphopenia and ECD were the two independent risk factors for OI, hence the need for customized immunosuppressive regimen in such transplant candidates. BKVN incidence remained high with a clear negative impact on allograft survival. In low-risk recipients, m-TOR based immunosuppressive therapy is the only prophylaxis to prevent BKVN and should be considered more widely. Two more issues need to be further studied: the specific role of pre-transplantation leucocytes subpopulation especially naive T-cells, and the difference between OI and common infections which have been described as the main cause of patient death after kidney transplantation.

## Figures and Tables

**Figure 1 jcm-08-00594-f001:**
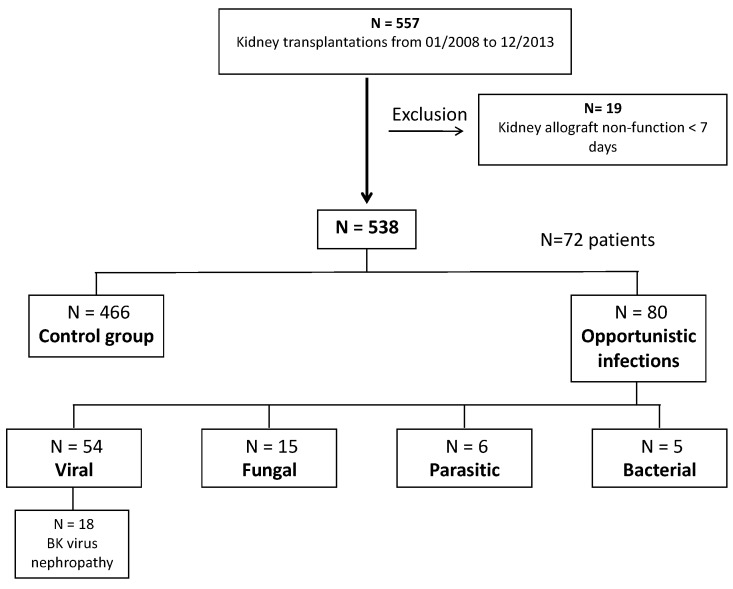
Flow chart of the study population. Between January 2008 and December 2013, 557 kidney transplantations were performed in *n* = 557 patients. Nineteen patients were excluded because of primary allograft non-function within the first week after transplantation. The final cohort included 538 transplantations in 538 patients.

**Figure 2 jcm-08-00594-f002:**
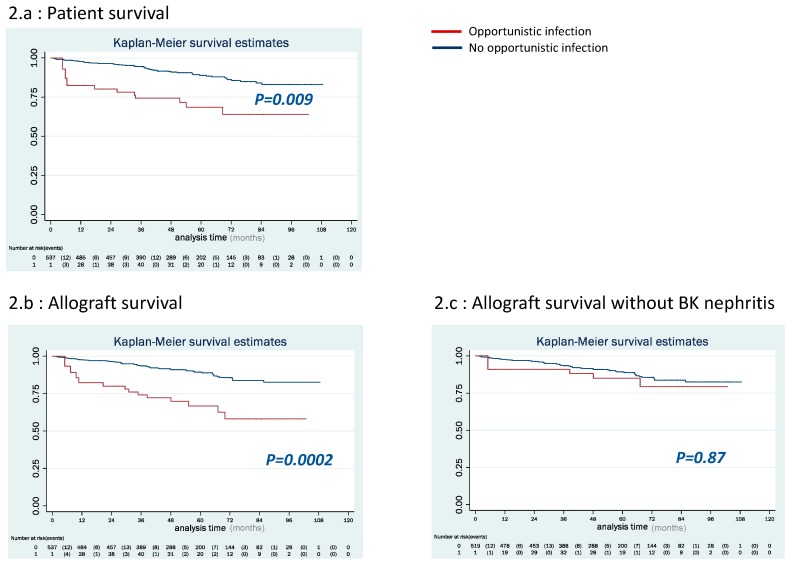
Patient, allograft, and event-free survival in both groups (Kaplan–Meier survival analysis): (**a**) in OI group, patient survival was significantly lower than in control group (*p* = 0.009); (**b**) allograft survival was significantly lower in OI group (*p* = 0.0002); and (**c**) allograft survival without BK virus nephropathy was not significantly lower in OI group (*p* = 0.87).

**Table 1 jcm-08-00594-t001:** Baseline characteristics of the patients included in the study.

Variables	Whole Cohort	Opportunistic Infections Group	Control Group	*p*-Value
	*n* = 538 (100%)	*n* = 72 (13%)	*n* = 466 (87%)	
**Recipients characteristics**				
Age, years, mean (SD)	52 ± 14	55 ± 15	51 ± 13	0.06
Sex, Female, *n* (%)	200 (37)	25 (35)	175 (38)	0.64
Initial nephropathy				
Glomerulopathy, *n* (%)	140 (26)	17 (24)	123 (26)	0.62
Unknown, *n* (%)	104 (19)	15 (21)	89 (19)	0.73
Diabetes Mellitus, *n* (%)	88 (16)	12 (17)	76 (16)	0.94
Hypertension, *n* (%)	54 (10)	8 (11)	46 (10)	0.75
Chronic interstitial nephropathy, *n* (%)	32 (6)	7 (10)	25 (5)	0.15
Genetic, *n* (%)	78 (15)	6(8)	72 (15)	0.11
Urologic, *n* (%)	29 (5)	6 (8)	23 (5)	0.23
Other, *n* (%)	13 (3)	2(3)	12 (3)	1.00
Diabetes before transplantation, *n*(%)	120 (22)	16 (22)	104 (22)	0.97
Dialysis, *n* (%)	488 (91)	67 (93)	421 (90)	0.46
Hemodialysis, *n* (%)	448 (83)	387 (83)	61 (85)	0.72
HIV +, *n* (%)	25 (5)	4 (6)	21 (5)	0.69
HCV +, *n* (%)	38 (7)	5 (7)	33 (7)	0.97
CMV +, *n* (%)	443 (82)	58 (81)	385 (83)	0.67
**Donor characteristics**				
Living donor, *n* (%)	43 (8)	3 (4)	40 (9)	0.20
Extended criteria donor, *n* (%)	245 (46)	48 (67)	197 (42)	0.0001
Age, years, mean (SD)	55 ± 16	59 ± 14	54 ± 16	0.02
eGFR, mL/min/1.73 m², median (IQR)	80 (58–103)	72 (56–94)	81 (58–103)	0.26
CMV +, *n* (%)	291 (54)	39 (54)	252 (54)	0.99
**Sensitization risk factors**				
Former kidney transplantation, *n* (%)	39 (7)	6 (8)	33 (7)	0.70
Anti-HLA antibodies, *n* (%)	285 (53)	38 (53)	247 (53)	0.93
Donor specific anti-HLA antibodies, *n* (%)	77 (14)	7 (10)	70 (15)	0.23
**Kidney transplant characteristics**				
Cold ischemia time, hours, median (IQR)	16 (12–20)	16 (13–20)	16 (12–20)	0.50
Immunosuppressive therapy				
Induction, *n* (%)	521 (97)	72 (100)	449 (96)	0.10
Basiliximab, *n* (%)	265 (49)	39 (54)	226 (48)	0.37
Antithymocyte globulin, *n* (%)	257 (48)	34 (47)	223 (48)	0.92
Rituximab, *n* (%)	47 (9)	6 (8)	41 (9)	0.90
Intravenous immunoglobulins, *n* (%)	89 (16)	9 (13)	80 (17)	0.32
Maintenance				
Calcineurin inhibitors, *n* (%)	534 (99)	72 (100)	462 (99)	0.43
Mycophenolate mofetil, *n* (%)	538 (100)	72 (100)	466 (100)	1.00
Steroids, *n* (%)	537 (99,8)	71 (99)	466 (100)	0.13
Belatacept	6 (1)	0 (0)	6 (1)	0.33
**Combined transplant**				
Heart, *n* (%)	4 (1)	1 (1)	3 (1)	0.67
Pancreas, *n* (%)	10 (2)	1 (1)	9 (2)	0.67
Liver, *n* (%)	19 (3)	4 (6)	15 (3)	0.67
**White blood cells at the time of transplantation**				
Leucocytes (G/L), median (IQR)	6.3 (5.2–7.9)	6.3 (5.3–8.2)	6.3 (5.2–7.8)	0.66
Neutrophils (G/L), median (IQR)	4.2 (3.1–5.4)	4.6 (3.2–6.2)	4.1 (3.0–5.4)	0.17
Lymphocytes (G/L), median (IQR)	1.3 (1.0–1.7)	1.2 (0.9–1.6)	1.3 (1.0–1.8)	0.04
CD4 T-cells (/µL), median (IQR)	525 (373–704)	493 (340–637)	526 (389–704)	0.15
CD4 T-cells (%), median (IQR)	45.1 (37.8–52.9)	46.1 (37.2–52.3)	45.1 (38.0–53.0)	0.51
CD8 T-cells (/µL), median (IQR)	312 (200–451)	284 (242–411)	314 (198–466)	0.79
CD8 T-cells (%), median (IQR)	26.9 (20.3–33.0)	27.7 (23.6–36.7)	26.6 (19.8–32.7)	0.11

**Table 2 jcm-08-00594-t002:** Follow-up of the patients included in the study.

Variables	Whole Cohort	Opportunistic Infections Group	Control Group	*p*-Value
	*n* = 538 (100%)	*n* = 72 (13%)	*n* = 466 (87%)	
**New onset diabetes after transplantation, *n* (%)**	34 (6)	5 (7)	29 (6)	0.82
**Acute rejection, *n* (%)**	136 (25)	23 (32)	113 (24)	0.19
T-cell mediated, *n* (%)	87 (16)	15 (21)	72 (16)	0.54
Antibody-mediated, *n* (%)	34 (6)	6 (8)	28 (6)	0.54
Mixed, *n* (%)	15 (3)	2 (3)	13 (3)	0.54
Time from transplantation, months (median, IQR)	5 (2–18)	7 (3–35)	4 (2–14)	0.14
Before opportunistic infection, *n* (%)	127 (24)	15 (21)	112 (24)	0.66
**Viral Infections**				
BK viruria	163 (30)	26 (36)	137 (29)	0.25
Time from transplantation, months (median, IQR)	6 (3–17)	7 (3–12)	6 (3–21)	0.83
Before opportunistic infection, *n* (%)	149 (28)	16 (22)	133 (28)	0.32
BK viremia	58 (11)	22 (31)	36 (8)	0.0001
Time from transplantation, months (median, IQR)	5 (3–8)	6 (3–12)	4 (3–6)	0.14
Before opportunistic infection, *n* (%)				
CMV viremia	178 (33)	30 (42)	148 (32)	0.10
Time from transplantation, CMV viremia (months, median, IQR)	4 (2–7)	5 (2–11)	3 (2–7)	0.22
Before opportunistic infection, *n* (%)	161 (30)	17 (24)	144 (31)	0.27
**12-month follow-up**				
eGFR mL/min/1.73 m² (median, IQR)	48 (36–60)	41 (31–53)	48 (37–61)	0.003
Allograft loss, *n* (%)	19 (4)	5 (7)	14 (3)	0.09
Time from transplantation, months (median, IQR)	7 (5–11)	10 (7–11)	6 (3–10)	0.252
Death, *n* (%)	16 (3)	3 (4)	13 (3)	0.52
Time from transplantation, months (median, IQR)	6 (2–9)	6 (5–7)	5 (2–9)	0.95
**Last follow-up**				
Time from transplantation, months (median, IQR)	52 (38–75)	48 (34–68)	53 (38–77)	0.045
eGFR mL/min/1.73 m² (median, IQR)	45 (36–60)	38 (29–52)	46 (34–62)	0.0009
Allograft loss, *n* (%)	68 (13)	13 (18)	55 (12)	0.14
Time from transplantation, months (median, IQR)	34 (13–54)	31 (110–48)	34 (14–55)	0.81
Death, *n* (%)	65 (12)	10 (14)	55 (12)	0.64
Time from transplantation, months (median, IQR)	34 (13–51)	29.9 (6–51)	34 (14–55)	0.58

**Table 3 jcm-08-00594-t003:** Opportunistic infection risk factors, univariate analysis.

	Whole Cohort			With No BK Virus Nephropathy		
Variables	HR	95% CI	*p*-Value	HR	95% CI	*p*-Value
**Recipient characteristics**						
Female	0.88	0.54–1.43	0.60	0.97	0.56–1.69	0.91
Age at transplantation	1.02	1.00–1.04	0.03	1.03	1.01–1.06	0.003
Dialysis	1.76	0.64–4.83	0.27	2.71	0.66–1.11	0.17
Hemodialysis	1.21	0.62–2.37	0.57	1.32	0.59–2.92	0.50
HIV+	0.95	0.30–3.03	0.94	1.30	0.41–4.18	0.66
CMV+	0.90	0.50–1.61	0.72	0.95	0.48–1.89	0.88
HCV+	1.01	0.41–2.51	0.98	1.09	0.39–3.01	0.87
Initial nephropathy						
Hypertension	1.11	0.53–2.31	0.78	1.53	0.72–3.25	0.27
Unknown origin	1.13	0.64–2.00	0.68	1.23	0.65–2.35	0.52
Diabetes	1.11	0.60–2.07	0.73	1.00	0.47–2.11	0.99
Genetic	0.48	0.21–1.12	0.09	0.54	0.21–1.35	0.19
Glomerulopathy	0.91	0.53–1.57	0.74	0.76	0.39–1.47	0.41
Tubular and interstitial	1.64	0.71–3.78	0.25	1.85	0.74–4.65	0.19
Urologic	1.56	0.68–3.60	0.30	1.39	0.50–3.86	0.52
Other	1.03	0.25–4.20	0.97	*	*	*
Diabetes before transplantation	1.02	0.57–1.81	0.95	0.91	0.45–1.82	0.78
**Donor characteristics**						
Age	1.02	1.01–1.04	0.01	1.03	1.01–1.05	0.002
Expanded Criteria Donor	2.76	1.68–4.54	<0.0001	3.74	2.03–6.90	<0.0001
eGFR	1.00	0.99–1.00	0.31	1.00	0.99–1.00	0.41
Living donor	0.47	0.15–1.49	0.20	0.20	0.03–1.48	0.12
CMV+	1.00	0.63–1.60	0.99	1.05	0.61–1.80	0.86
**Sensitization risk factors**						
Anti HLA antibodies	0.99	0.62–1.59	0.98	0.97	0.56–1.66	0.90
Former kidney transplantation	1.02	0.41–2.54	0.96	1.10	0.40–3.04	0.86
Donor specific anti-HLA antibodies	0.69	0.32–1.51	0.36	0.39	0.12–1.25	0.11
**Combined Transplants**						
Pancreas	0.69	0.10–5.00	0.72	0.92	0.13–6.65	0.93
Liver	1.87	0.68–5/12	0.23	1.90	0.59–6.09	0.28
Heart	*	*	*	*	*	*
**Kidney Transplant Characteristics**						
Cold ischemia time	1.02	0.99–1.05	0.28	1.03	0.99–1.07	0.18
Induction Immunosuppressive regimen						
Basiliximab	1.21	0.76–1.93	0.43	1.51	0.87–2.61	0.15
Antithymocyte globulin	1.01	0.63–1.61	0.96	0.84	0.48–1.45	0.52
Intravenous Immunoglobulin	0.74	0.37–1.48	0.39	0.42	0.15–1.16	0.09
Rituximab	1.03	0.44–2.37	0.95	0.45	0.11–1.83	0.26
Maintenance immunosuppressive regimen						
Calcineurin inhibitors	*	*	*	*	*	*
Mycophenolate Mophetil	*	*	*	*	*	*
Steroids	*	*	*	*	*	*
Belatacept	*	*	*	*	*	*
**White blood cells at the time of transplantation**						
Leucocytes (/1000)	1.01	0.92–1.11	0.84			
Neutrophils (/1000)	1.05	0.95–1.15	0.37			
Lymphocytes (/1000)	0.61	0.40–0.95	0.028			
TCD4 cells (/1000)	0.34	0.08–1.38	0.13			
TCD4 cells (%)	0.98	0.96–1.01	0.25			
TCD8 cells (/1000)	1.28	0.31–5.24	0.73			
TCD8 cells (%)	1.02	1.00–1.05	0.06			
**White blood cells as time-varying variables during follow-up**						
Leucocytes (/1000)	0.94	0.84–1.05	0.28	0.89	0.78–1.02	0.09
Neutrophils (/1000)	0.98	0.87–1.10	0.72	0.92	0.79–1.07	0.28
Lymphocytes (/1000)	0.74	0.50–1.11	0.15	0.70	0.43–1.13	0.14
TCD4 cells (/1000)	0.31	0.11–0.83	0.02	0.45	0.15–1.33	0.15
TCD4 cells (%)	0.97	0.95–0.99	0.004	0.98	0.96–1.00	0.11
TCD8 cells (/1000)	0.70	0.23–2.10	0.52	0.90	0.27–2.98	0.86
TCD8 cells (%)	1.01	0.99–1.03	0.42	1.00	0.98–1.02	0.94
**Follow up**						
CMV viremia before OI	1.29	0.76–2.18	0.34	1.53	0.85–2.76	0.16
BK viruria before OI	1.94	1.15–3.28	0.01	0.34	0.12–0.96	0.04
Acute rejection before OI	1.48	0.83–2.63	0.19	1.38	0.70–2.71	0.35
High calcineurin inhibitors level in the month before OI	2.06	0.97–4.35	0.06	3.87	1.53–9.80	0.004
Mycophenolate Mofetil at the time of OI	1.134	0.45–2.83	0.79	0.41	0.10–1.78	0.24
mTOR at the time of OI	0.49	0.15–1.56	0.23	0.56	0.17–1.80	0.33

* data non analyzed.

**Table 4 jcm-08-00594-t004:** Patient and allograft survival independent risk factors (time varying Cox model).

**Patient Overall Survival**			
**Variables**	**HR**	**95% CI**	***p*-Value**
Recipient age at transplantation	1.08	1.05–1.11	<0.0001
TCD8 cells (/1000), time-varying during follow-up	0.26	0.07–1.01	0.052
Neutrophils (/1000), time-varying during follow-up	1.12	0.99–1.27	0.08
HCV+	3.02	1.39–6.55	0.005
Former kidney transplantation	3.18	1.35–7.50	0.008
Diabetes	1.83	1.04–3.22	0.04
**Allograft survival**			
**Variables**	**HR**	**95% CI**	***p*-Value**
Donor age	1.02	1.00–1.04	0.03
TCD4 cells (/1000), time-varying during follow-up	0.23	0.08–0.67	0.007
Acute rejection before opportunistic infection	3.28	1.93–5.57	<0.0001
Opportunistic infection episode	2.53	1.29–4.95	0.007
Donor specific anti-HLA antibodies before transplantation	1.92	0.97–3.78	0.06
CMV+ donor	1.83	1.05–3.19	0.03
Diabetes	1.97	1.15–3.39	0.014

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
