# Peer review of "Epidemiology, Risk Factors, and Outcomes of Opportunistic Infections after Kidney Allograft Transplantation in the Era of Modern Immunosuppression: A Monocentric Cohort Study"

_jcm, 2019, doi:10.3390/jcm8050594_

Round 1
Reviewer 1 Report
The authors attempt to describe their findings regarding opportunistic infection after kidney allograft transplantation over a 5 year period. While the concept, findings and conclusion are interesting, the paper lack clarity especially with the introduction and methods sections. Overall, the manuscript is also poorly written with several grammatic errors making it difficult at times to appreciate or understand what the authors are trying to pass across.
Suggest:
Manuscript revision by a professional English editing servicing with attention to sentence structure and grammar.
Extended criteria donor was one of the independent risk factors for opportunistic infection described by authors but this term was never described in the paper. The authors should briefly describe this term in addition to including relevant reference(s).
Authors reported very low CMV infections but did not describe the type of CMV prophylaxis employed by center (universal versus pre-emptive). This should be addressed.
Authors reported under "Materials and Methods" Opportunistic infection section that 'all episodes were retrospectively and blindly validated by an Infectious diseases specialist'. Authors should describe further how blinding was done. Were these specialist part of the study group?
Under the result section, authors seemed to equate syndrome with pathogens; listing Kaposi sarcoma and BKV nephropathy as viral opportunistic infections. These should be better clarified.
The authors did not describe any limitation(s) encountered with the study. This should be included.
Author Response
The authors attempt to describe their findings regarding opportunistic infection after kidney allograft transplantation over a 5 years period. While the concept, findings and conclusion are interesting, the paper lack clarity especially with the introduction and methods sections. Overall, the manuscript is also poorly written with several grammatic errors making it difficult at times to appreciate or understand what the authors are trying to pass across.
Point 1: Manuscript revision by a professional English editing servicing with attention to sentence structure and grammar.
Response 1: As suggested by Reviewer 1, a professional English editing service reviewed our paper extensively especially sentence structure and grammar. We have no doubt that the review has enhanced the quality of our manuscript.
Point 2: Extended criteria donor was one of the independent risk factors for opportunistic infection described by authors but this term was never described in the paper. The authors should briefly describe this term in addition to including relevant reference(s).
Response 2: We added in methods the definition of extended criteria donor with one relevant reference (Port Transplantation 2002). Extended criteria donors are donors older than 60 years or between 50 and 60 years, with two of the three following criteria (i) hypertension, (ii) pre-retrieval serum creatinine > 1.50 mg/dL and (iii) cerebrovascular cause of brain death.
Point 3: Authors reported very low CMV infections but did not describe the type of CMV prophylaxis employed by the center (universal versus pre-emptive). This should be addressed.
Response 3: As requested by Reviewer 1, we added in methods, the CMV prophylaxis currently in use in our center. The management for CMV prophylaxis follows international recommendations (Kotton et al Transplantation 2013). Prophylaxis involves the administration of oral valganciclovir to high (donor positive serology and recipient negative serology) and intermediate (recipient positive serology treated with thymoglobulin) risk patients. Duration of prophylaxis was 6 months in high risk patients and 3 months in intermediate ones.
Point 4: Authors reported under "Materials and Methods" Opportunistic infection section that 'all episodes were retrospectively and blindly validated by an Infectious diseases specialist'. Authors should describe further how blinding was done. Were these specialist part of the study group?
Response 4: The infectious diseases specialist is part of the study group. All medical reports (clinical and biological data) of infections that happened in kidney-transplant recipients included in the study were reviewed without the patient name and the final conclusion. We added these details in the methods section to clarify the opportunistic infection selection.
Point 5 : Under the result section, authors seemed to equate syndrome with pathogens; listing Kaposi sarcoma and BKV nephropathy as viral opportunistic infections. These should be better clarified.
Response 5: Considering BKV nephropathy, we decided to include it in opportunistic infections because BK virus is highly seroprevalent in humans but appear to cause clinical disease only in immunocompromised patients and almost all after kidney transplantation. Clinical disease includes tubulointerstitial nephritis called BKV-induced nephropathy and ureteral stenosis (Nankivell American Journal of Transplantation 2017 2065)). Development of BKV nephropathy is directly related to plasma viral load (Nankivell American Journal of Transplantation 2017 2065).
Considering Kaposi sarcoma, it is classified in four types based on the clinical circumstances in which it develops: classic, endemic, organ transplant-associated and epidemic or AIDS-related. The first two types are not associated with immune deficiency. The third one associated with organ transplantation is similar to epidemic Kaposi Sarcoma in its clinical manifestations and usually regresses with reduction in immunosuppression (Penn Transplantation 1997 669).
We added in the manuscript what justifies that these two infections are opportunistic infections.
Point 6 : The authors did not describe any limitation(s) encountered with the study. This should be included.
Response 6: As asked by reviewer 1, we added limitations of our study in the discussion. The first one is being a single center study and retrospective. These results must be confirmed in a prospective multicentric cohort. However, the single center study implies only one way to manage immunosuppression after transplantation. The second one is that we performed an overview of opportunistic infections without considering specific prognosis of each infection. We added those limits in the discussion section.

Reviewer 2 Report
This is a retrospective, single center study on opportunistic infections (OI) after kidney transplantation.
The paper describes well the current status of opportunistic infections when prophylaxis and intensive monitoring strategies for OI including CMV, PCP and BKV are recommended by international guidelines.
However, some results need to be clarified by the authors and there is also an over emphasis on the interpretation of the results.
Effect of OIs on allograft survival was mostly due to BKV nephropathy. So author's conclusion was over emphasized.
Authors described ECD was an independent risk factor for OI. But use of induction therapy or degree of immune suppressive medication did not affect the development of OI. Can authors explain why there are higher incidence of OIs in ECD, if degree of immune suppression did not affect the OIs?
In table 2, median time of BK viruria was 7 months, however median time of BK viremia was 6 months in OI group. This tendency was similar in control group (6 vs. 4 months).
BK viruria precedes the viremia and there is a window period.
What was the BKV monitoring strategy applied to these patients?
Authors described the blood lymphocyte at the time of transplantation was one of the risk factors, but number and percentage of lymphocyte after transplantation were not different between OI and control group.
Authors already described that OIs mostly occurred beyond one year after transplantation. Can authors explain how lymphocyte count can affect the lately developed OIs?
In figure 2. authors showed the OIs episodes had effect on patient's survival. But they omitted OIs as variables for patients overall survival in table 4 and described that OS was not an independent risk factor of death after multivariable analysis.
Please make sure whether OIs was included or not.
Author Response
This is a retrospective, single center study on opportunistic infections (OI) after kidney transplantation.
The paper describes well the current status of opportunistic infections when prophylaxis and intensive monitoring strategies for OI including CMV, PCP and BKV are recommended by international guidelines.
However, some results need to be clarified by the authors and there is also an over emphasis on the interpretation of the results.
Point 1 : Effect of OIs on allograft survival was mostly due to BKV nephropathy. So author's conclusion was over emphasized.
Response 1: As suggested by Reviewer 2, we added in the conclusion that impact of OIs on allograft survival was mostly due to BKV nephropathy.
Point 2 : Authors described ECD was an independent risk factor for OI. But use of induction therapy or degree of immune suppressive medication did not affect the development of OI. Can authors explain why there are higher incidence of OIs in ECD, if degree of immune suppression did not affect the OIs?
Response 2: Actually, incidence of OIs was significantly higher in ECD recipients. In France, recipient age is significantly correlated with donor age as it is a one of the criteria to allocate a kidney allograft to a recipient. Moreover, in kidney allograft recipients, older donor age, independently of recipient age, increases the rate of acute allograft rejection and infections irrespective of immune suppression therapy. We think that the underlying immune system is more important than immunosuppressive therapy.
Point 3 : In table 2, median time of BK viruria was 7 months, however median time of BK viremia was 6 months in OI group. This tendency was similar in control group (6 vs. 4 months). BK viruria precedes the viremia and there is a window period. What was the BKV monitoring strategy applied to these patients?
Response 3: During the first year after kidney transplantation, BK viruria tests were performed at one, two, three, six, nine, and 12 months. BK viremia was checked once BK viruria was positive. If BK viruria (associated with BK viremia or not) was positive, a blood test was performed every two weeks. We added these details in methods.
Point 4 : Authors described the blood lymphocyte at the time of transplantation was one of the risk factors, but number and percentage of lymphocyte after transplantation were not different between OI and control group.
A statistically significant difference was indeed found in blood lymphocytes count at the time of transplantation between controls and patients with OI in univariate analysis (Table 1 – [OI] Median 1.2 (IQR 0.9-1.6) vs. [Controls] 1.3 (1.0-1.8); p=0.04), a result confirmed in time-to-event analysis (Univariate analysis: HR 0.61 [0.40-0.95]; multivariate analysis: HR 0.60 [0.38-0.94]; P=0.026). However and as noted by the reviewer, no statistically significant association was found between CD4/ CD8 numbers (%) at the time of transplant and OI status (either comparing controls to OI patients [Table 1] or using time-to-event analysis [Table 3]) but CD4 during follow-up was found to be an independent protective factor in time-to-event multivariate analysis (HR 0.98 [0.96-0.99], P=0.015).
It is therefore possible that the total count in lymphocytes had a superior predictive value for OI than the separate levels of CD4/CD8. In addition, it should be noticed that study population for analyses on CD4/CD8 was slightly decreased due to missing information on these variables (N=456 available out of 538), thus possibly resulting in a moderate loss of statistical power.
Point 5 : Authors already described that OIs mostly occurred beyond one year after transplantation. Can authors explain how lymphocyte count can affect the lately developed OIs?
Response 5: Wide use of prophylaxis like Cotrimoxazole and oral Valganciclovir decreased the incidence of PCP, Nocardia, and CMV disease during the first year after kidney transplantation. We supposed that this could be a cumulative effect of immunosuppressive therapies on older lymphopenic patients.
Point 6 : In figure 2. authors showed the OIs episodes had effect on patient's survival. But they omitted OIs as variables for patients overall survival in table 4 and described that OS was not an independent risk factor of death after multivariable analysis.
Please make sure whether OIs was included or not.
We confirm that OI was a significant predictor for overall survival in univariate analysis only (as shown in Figure 2, log-rank p=0.009), but OI lost its statistical significance after multivariable adjustment for recipient age at transplantation, TCD8 cells (/1000) during follow-up, neutrophils (/1000) during follow-up, HCV+ status, former kidney transplantation and diabetes. As a consequence and in accordance with our statistical analysis strategy detailed in the methods section, OI was left out of Table 4 showing only results from the final multivariable model after a stepwise backward approach was applied.

Round 2
Reviewer 2 Report
The authors' reply to the opinion of the reviewer is reasonable and the paper has been modified favorably overall.